applied mathematics/complexity/mathematical modelling

scaling, road accidents, urban systems

**Author for correspondence:**
C. Cabrera-Arnau
e-mail: c.cabrera-arnau@ucl.ac.uk

# Uncovering the behaviour of road accidents in urban areas

## C. Cabrera-Arnau[1], R. Prieto Curiel[2] and S. R. Bishop[1]

[1]Department of Mathematics, University College London, Gower Street, WC1E 6BT London, UK
[2]Research in Spatial Economics (RiSE-group), Department of Mathematical Sciences, Universidad EAFIT, Medellin, Colombia

CC-A, 0000-0002-2732-6436

Different patterns in the incidence of road accidents are revealed when considering areas with increased levels of urbanization. To understand these patterns, road accident data from England and Wales is explored. In particular, the data are used to (i) generate time series for comparison of the incidence of road accidents in urban as opposed to rural areas, (ii) analyse the relationship between the number of road accidents and the population size of a set of urban areas, and (iii) model the likelihood of suffering an accident in an urban area and its dependence with population size. It is observed that minor and serious accidents are more frequent in urban areas, whereas fatal accidents are more likely in rural areas. It is also shown that, generally, the number of accidents in an urban area depends on population size superlinearly, with this superlinear behaviour becoming stronger for lower degrees of severity. Finally, given an accident in an urban area, the probability that the accident is fatal or serious decreases with population size and the probability that it is minor, increases sublinearly. These findings promote the question as to why such behaviours exist, the answer to which will lead to more sustainable urban policies.

## 1. Introduction

### 1.1. Road accidents and urban areas: the current picture

The Global Status Report on Road Safety 2018 published by the World Health Organization states that more than 1.35 million people die each year on the world's roads [1]. The number of fatalities relative to the size of the world's population has stabilized in recent years. However, if this trend is maintained, the sustainable development goals (SDGs) target 3.6 [2] to reduce, by half, road traffic deaths by 2020 will not be met. Road accidents pose a serious problem for the economy, especially in low- and middle-income countries, where the death rates due to road injuries are three times higher than in high-income countries. At a worldwide level, road accidents are the leading cause for deaths among young people aged between

5 and 29, and the eighth cause for all the age groups, above HIV/AIDS, tuberculosis and diarrhoeal diseases. In addition to deaths on the roads, about 50 million people suffer non-fatal road injuries as well as other indirect health consequences each year [1].

In recent decades, the number of motor vehicles in the world has risen from 0.85 billion in the year 2000 to 2.1 billion in 2016 [1], leading to an increased exposure to traffic for most people. This motorization has grown hand in hand with urbanization. Since 1950, the urban population of the world has rapidly increased, so that in 2018, 55% of the world's population live in urban areas [3]. However, the percentage of urban population varies from country to country being as much as 82% in Northern America or 74% in Europe [3]. This change, to a society of city dwellers, will have a significant, but still poorly understood, impact on the global environment that transcends urban boundaries [4]. The quantitative understanding of urban organization and dynamics is thus key for a successful transition to sustainability [5].

This paper investigates some of the effects that the rapid urbanization that the world is undergoing has on the incidence of road accidents. Road safety data from England and Wales (E&W) corresponding to the period spanning from 2008 to 2018 are chosen to aid with the study due to its accuracy and accessibility. Furthermore, the application of laws in line with the so-called 'best practice' on behavioural risk factors—such as speed, drink-driving and failing to use motorcycle helmets, seat-belts and child restraints—have positioned the UK among the top five countries with regards to the lowest mortality rates on the roads. With the aim of reducing to zero the number of people killed or seriously injured on the roads, the UK also incorporates a safety strategy called Vision Zero https://www.trafikverket.se/en/startpage/operations/Operations-road/vision-zero-academy/, which was initiated in Sweden in the 1990s, and since then has been adopted by numerous countries in the framework of the Safe System approach towards road safety. A more detailed description of the datasets and methods used in this paper can be found in §2. This data is then used for different purposes: in §3.1, time series are generated in order to explore and compare the general behaviour of road accidents of different degrees of severity in both urban and rural areas; in §3.2, the dependence of the number of road accidents in urban areas with the population size of these urban areas is mathematically formulated and contrasted with the data; in §3.3, a model of how the likelihood of suffering an accident in an urban area varies with population size is proposed. Finally, §§4 and 5 provide a discussion and conclusions regarding the observed behaviours.

The simplicity of the mathematical treatment in both the modelling and the analysis of the dataset used in this paper makes it accessible to the wider community. Therefore, the results and conclusions reached here will be of value to anyone working to achieve the SDGs [2], in particular, Goal 3 (ensure healthy lives and promote well-being for all at all ages) and Goal 11 (make cities and human settlements inclusive, safe, resilient and sustainable).

## 1.2. Relationship between road accident incidence and population size

Despite the fact that road accidents are a global concern, there are still unanswered questions about how the number of accidents in urban areas scales with the population size or the population density of a given location. When it comes to quantifying different aspects of cities, simple *per capita* measures are most commonly used. However, these assume implicitly that urban characteristics increase linearly with population size. This assumption is not entirely correct since it ignores the inherent nonlinear nature of the organization and dynamics of cities with different population sizes. As Bettencourt *et al.* observe in [6], these nonlinearities are manifested as scaling laws which show that urban areas display the emergent phenomenon of agglomeration, so that if $P$ represents the population size of the urban areas under consideration and $I$ is an indicator of some sort, then

$$I(P) = \alpha P^{\beta}, \tag{1.1}$$

where the scaling exponent $\beta$ is, in general, different from 1, and $\alpha$ is a proportionality constant. For example, assuming this form of model to fit to data, it has been found that the indicator for economic productivity varies with population size according to $\beta = 1.15$ [6], i.e. it increases systematically 2.22 times with every doubling of an urban settlement's population. Similarly, the walking speed [7], the criminal activity [8], the $CO_2$ emissions [9], the average number of contacts and communication activity [10], the economic diversification [11], the road length distribution [12], the number of people migrating to a city [13], the amount of coverage received from the media [14] and the number tweets produced in a city [15] have all been found to scale as a power law with population size.

**Table 1.** Number of accidents during the period 2008–2018 in England and Wales according to their severity and the type of area where they occurred.

| | urban | | | rural | | | |
|---|---|---|---|---|---|---|---|
| | fatal | serious | minor | fatal | serious | minor | total |
| 2008 | 773 | 12 257 | 88 896 | 1321 | 8628 | 46 380 | 158 255 |
| 2009 | 686 | 11 609 | 85 997 | 1172 | 8396 | 43 965 | 151 825 |
| 2010 | 539 | 11 067 | 82 369 | 1003 | 7666 | 41 463 | 144 107 |
| 2011 | 584 | 11 634 | 81 442 | 1037 | 7681 | 39 119 | 141 497 |
| 2012 | 570 | 11 757 | 77 547 | 911 | 7427 | 37 682 | 135 894 |
| 2013 | 493 | 10 716 | 73 044 | 951 | 7383 | 36 309 | 128 896 |
| 2014 | 550 | 11 198 | 79 090 | 926 | 7937 | 37 344 | 137 045 |
| 2015 | 552 | 10 870 | 75 546 | 910 | 7700 | 35 569 | 131 147 |
| 2016 | 536 | 11 751 | 71 573 | 957 | 8544 | 34 880 | 128 268 |
| 2017 | 581 | 13 120 | 69 284 | 954 | 8037 | 30 872 | 122 848 |
| 2018 | 592 | 13 533 | 64 561 | 927 | 8251 | 28 317 | 116 181 |
| totals | 6483 | 129 512 | 849 349 | 11 069 | 87 650 | 411 900 | 1 495 963 |

In 1949, before the application of power laws such as in equation (1.1) became popularized, Smeed [16] proposed a rule of the following form:

$$\frac{D}{P} = \alpha \left(\frac{N}{P}\right)^{\beta},$$

(1.2)

that relates the yearly number of deaths by accidents $D$, the population $P$ and the number of registered vehicles $N$ in a certain country. When data referring to different countries was considered, the parameters in (1.2) were estimated to be, approximately, $\alpha \approx 0.0003$ and $\beta \approx 1/3$. This rule, known as Smeed's Law, was revised by Adams in 1987 [17], who proposed using vehicle miles instead of the number of vehicles per person as a measure of exposure to traffic. Smeed's Law was further revised by Andreassen in two publications from 1985 [18] and 1991 [19], where he discussed that the values of the parameters $\alpha$ and $\beta$ given by Smeed are not correct since they arise as the result of a spurious correlation of the variables $D/P$ and $N/P$, as both of these variables contain the population $P$ in the denominator. Andreassen also recommended considering the number of accidents and their severity instead of just the number of deaths, especially when the results are created for accident reduction purposes.

In the current work, a relationship of the type (1.1) between the number of accidents occurring in urban areas and the population size of their location is hypothesized. The estimated value of the scaling exponent $\beta$ is found for accidents of different degrees of severity by fitting equation (1.1) to the data instead of equation (1.2). Furthermore, based on the same mathematical formulation, an expression for the likelihood of suffering road accidents in an urban area of a certain population size during a given period of time is derived.

## 2. Data and methods

### 2.1. Accident data

The databases 'Road Safety Data—Accidents' corresponding to the years from 2008 to 2018 contain all the accidents that occurred in Great Britain (England, Wales and Scotland) over this period and can be found for download on the website https://data.gov.uk/dataset/cb7ae6f0-4be6-4935-9277-47e5ce24a11f/road-safety-data. Due to the fact that geographical data for E&W and for Scotland and Northern Ireland is held separately, only the road accidents that occurred in English and Welsh urban and rural roads during the period 2008–2018 are considered here. Table 1 gathers the number of accidents in E&W according to their severity (fatal, serious or minor) and the type of area (urban or

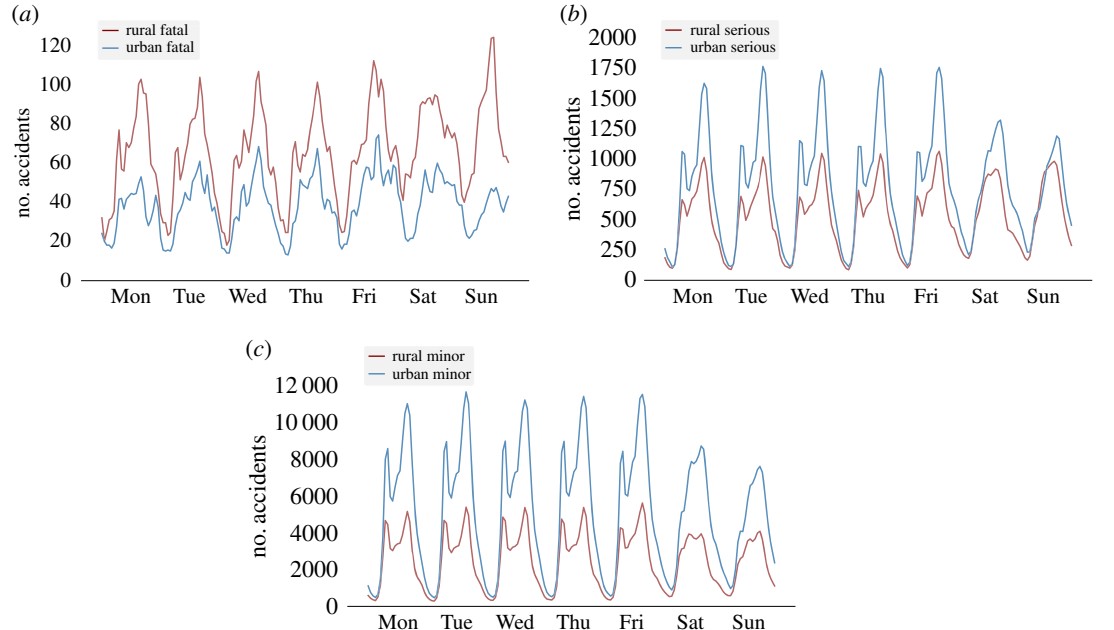

**Figure 1.** Time-series data for accidents of different degrees of severity in urban and rural areas of England and Wales over the years 2008–2018. (*a*) Fatal accidents, (*b*) serious accidents and (*c*) minor accidents.

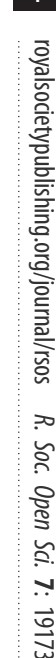

rural) where they occurred. The explanation of how this classification is reached can be found in §§A.1 and A.2 of the appendix.

## 2.2. Scaling model

The urban areas taken into account when fitting the data to the scaling equation (1.1) are the so-called built-up areas (BUAs) from E&W with more than 10 000 inhabitants. Further details on the precise definition of 'urban area' used here are given in §A.1 of the appendix.

It would be reasonable to think that higher population density might lead to more traffic and therefore, higher chances for the occurrence of accidents. There are reasons, however, to discard this hypothesis, since the BUAs considered include big urban areas such as Greater London (BUA code: E34004707), which has a relatively moderate average population density of 56 prs ha$^{-1}$ (as per the 2011 UK Census), and yet, registers a disproportionately large number of accidents. On the contrary, among these BUAs, there are also smaller ones that, despite their high population density, have a low number of accidents, as it is the case for New Addington (BUA code: E34000214), with 71 prs ha$^{-1}$.

For the purposes of this paper, given $n$ urban areas, their population size, denoted by $P_i$, with $i = 1, \ldots n$, is therefore assumed to be the only explanatory variable for the number of accidents. The count of the number of accidents throughout a given period of time in each of the urban areas under consideration is denoted by $A_i$. Assuming that $A_i$ follows a Poisson distribution, the expected value for the number of accidents occurring during the same period of time in the $i$th urban area is $\mathbb{E}[A_i] = \mu_i$. Thus, equation (1.1), rewritten in terms of these variables, reads as

$$\mu_i(P_i) = \alpha P_i^{\beta}, \tag{2.1}$$

where $\alpha$ and $\beta$ are the model parameters. The maximum-likelihood estimates for $\log \alpha$ and $\beta$ are obtained via a Poisson regression and their values are given in table 2.

# 3. Results

## 3.1. Accident incidence by area and degree of severity

A clearer understanding of the incidence of road accidents of different severity levels in urban areas and how this compares to the incidence in rural areas, can be gained by observation of time-series data. In figure 1, time series for accidents of fatal (*a*), serious (*b*) and minor (*c*) severity occurring in E&W from

**Table 2.** Maximum-likelihood estimates for the parameters corresponding to the power laws describing the relation between the number of accidents of different degrees of severity in urban areas of England and Wales from 2008 to 2018 and their population sizes.

|  | $\beta$ | $\log \alpha$ |
|---|---|---|
| fatal | $\beta_F = 1.085 \pm 0.006$ | $\log \alpha_F = -10.061 \pm 0.083$ |
| serious | $\beta_S = 1.075 \pm 0.001$ | $\log \alpha_S = -6.930 \pm 0.018$ |
| minor | $\beta_M = 1.120 \pm 0.001$ | $\log \alpha_M = -5.662 \pm 0.007$ |
| all severities | $\beta_T = 1.114 \pm 0.001$ | $\log \alpha_T = -5.428 \pm 0.007$ |

the year 2008 until the year 2018 are displayed. Accident frequency by the hour and day of the week is represented in the vertical axis. This discretization of the time dimension, represented in the horizontal axis, is chosen for display over others (for example, by day and month) as it produces more insightful time patterns. A simple 2-hour window moving mean calculation is applied.

The occurrence of minor accidents, in both rural and urban areas, follows a bimodal distribution with daily periodicity during the weekdays, with each of the peaks corresponding to the morning and evening rush hours. The larger size of the peaks corresponding to the evening rush hour is something to note, it might be due to factors such as a more congested traffic flow, higher levels of stress and exhaustion in the drivers after a day of work, reduced visibility during the evening or a combination of these. The distribution becomes closer to unimodal during the weekends, reflecting the fact that less people are travelling to work. Similar patterns are observed for serious accidents, although in rural areas, the trend is not as strong. Both minor and serious accidents have a much higher incidence in urban areas (figure 1*b*,*c*).

Based on figure 1*a*, rural areas in E&W seem to experience more fatalities than urban areas. Fatal accidents reach their frequency peak on Sundays in rural areas, when perhaps more people are travelling away from their usual place of residence in urban areas.

## 3.2. Scaling of accidents in England and Wales

The total number of accidents in urban areas of different population sizes from E&W can be assumed to follow a Poisson distribution. The data is then fitted using a Poisson regression to equation (2.1). The same statistical analysis is applied across all degrees of severity. The maximum-likelihood estimates for the parameters, with 95% confidence intervals, are displayed in table 2. These estimates are obtained with road accident data corresponding to the years 2008–2018 and the populations corresponding to the middle point of this period: the 2013 mid-year population estimates (see §A.1 for details on how the mid-year population estimates for the urban areas are computed). The maximum-likelihood estimates for the scaling exponents corresponding to the road accidents and populations of each individual year are gathered in table 3.

The results for the estimated values of the parameters suggest that, while the number of accidents increases with population size, it tends to increase faster for lower levels of severity. In other words, minor accidents in E&W present a strong superlinear behaviour whereby the number of accidents in urban areas grows faster than proportionally with population size. For serious and fatal accidents, the superlinear behaviour is weaker although the scaling exponent still remains greater than 1. These results are displayed in figure 2.

A sensitivity analysis for the values of the estimated scaling exponents $\beta_F$, $\beta_S$, $\beta_M$ and $\beta_T$ is provided in the appendix, §A.4. The analysis tests variations in the estimated values when different sets of urban areas are taken into account.

## 3.3. Probability of an accident of a given degree of severity

In the previous section, the number of accidents in urban areas is modelled as a power law of their population size. But, given an accident, does the probability that it is of minor, serious or fatal severity also vary with the population size of the urban areas?

Consider the probability mass function over the random variable $X$, which corresponds to the number of accidents that an individual residing in an urban area of population $P$ suffers in a given

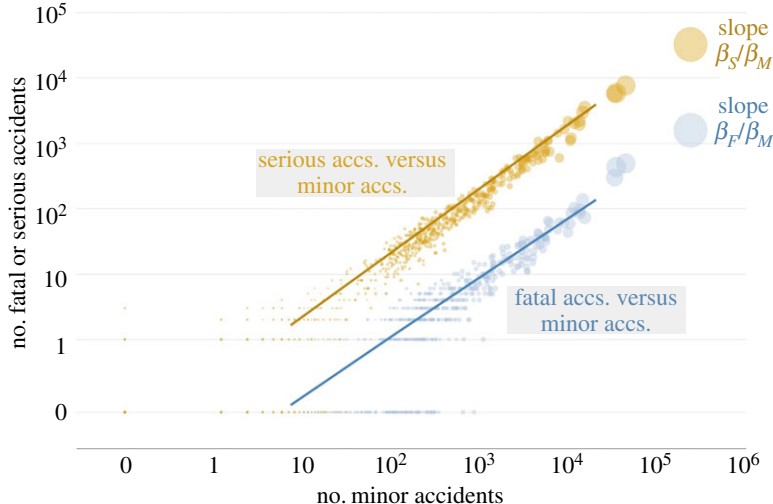

**Figure 2.** Number of serious (in yellow) and fatal (in blue) accidents versus number of minor accidents occurring in urban areas in England and Wales during the years 2008–2018. Each dot represents an urban area and the size of the dot represents the population size in the middle of this period of time, i.e. the mid-2013 population estimate. The lines are obtained with the best fit estimates for the parameters in equation (2.1).

period of time. If $A$ is the total number of accidents occurring in the urban area in the same period of time, this probability mass function is a binomial distribution of the type

$$\Pr(X = k) = \binom{A}{k} p^k (1 - p)^{A-k}, \tag{3.1}$$

where $p$ is the probability that a certain individual in the urban area under consideration suffers $k$ accidents. Since the population size is $P$, then, assuming all individuals have the same probability of suffering an accident, $p = 1/P$. According to equation (3.1), the probability that a given individual suffers at least one accident is given by

$$\Pr(X > 0) = 1 - \Pr(X = 0) = 1 - (1 - p)^A \approx Ap,$$

where the approximation in the last step can be taken when $p \ll 1$. For the urban areas taken into account in this paper, all the populations are above 10 000, so $p$ is smaller than or equal to $10^{-5}$, and therefore, the approximation is justified. Using equation (2.1) to write $A$ in terms of $P$ and the fact that $p = 1/P$

$$\Pr(X > 0) \approx \alpha_T P^{\beta_T - 1}. \tag{3.2}$$

This equation can also be applied to accidents of a particular degree of severity. It then follows that the probability that given an accident, the accident is, for example, fatal, is the conditional

$$\Pr(F|E) = \frac{\Pr(F \cap E)}{\Pr(E)} = \frac{\alpha_F}{\alpha_T} P^{\beta_F - \beta_T}, \tag{3.3}$$

where $E$ denotes the event of an accident and $F$, the event of a fatal accident. The probability that given an accident, the accident is serious or minor can be obtained analogously.

According to the estimated values for the parameters shown in table 2 from the dataset described in §2.1, the probability that an individual suffers at least a fatal accident (an individual is said to suffer a fatal accident if there was at least one fatal injury in the vehicle) in an urban area during the period 2008–2018 increases sublinearly with the population size of the urban area since the exponent in equation (3.2) for fatal accidents is $\beta_F - 1 = 0.085 \pm 0.006$. A similar result can be observed for the case of serious and minor accidents: the probability of suffering at least a serious or a minor accident increases sublinearly with the population size.

From equation (3.3), it can be learned that the probability that, given an accident in an urban area, the accident is fatal, decreases sublinearly with population size, since the exponent is $\beta_F - \beta_T = -0.029 \pm 0.006$. Therefore, even though the probability that suffering at least an accident during the period 2008–2018 becomes higher as the population size of an urban area increases, the probability that this

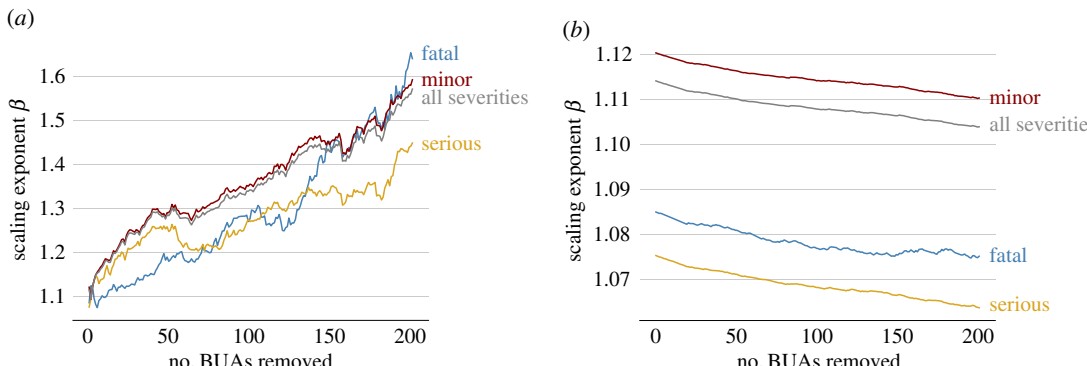

**Figure 3.** Sensitivity analysis of the estimated scaling exponents for accidents of different severity in urban areas from England and Wales during the years 2008–2018 when removing (a) the $k$th largest urban areas or (b) the $k'$th smallest urban areas, with $k$, $k' = 0, \ldots, 200$.

accident is fatal decreases with the population size. Serious accidents follow a similar trend. However, the probability that a given accident is minor increases sublinearly with population size as $\beta_M - \beta_T = 0.006 \pm 0.001$.

## 4. Discussion

Environmental factors and their influence on drivers' behaviour, combine to give rise to specific trends in the incidence of road accidents. Through the generation of time series regarding the frequency of road traffic accidents occurring in E&W from the year 2008 to the year 2018, it can be learned that accidents leading to serious and minor injuries occur more often on urban roads than on rural roads and they produce clear patterns throughout the week: on weekdays, there are two daily peaks of different height in the number of accidents in both urban and rural roads, the lower one corresponding to the morning rush hour, and the higher one, to the evening rush hour; at weekends, there are less accidents than on weekdays and the two peaks are no longer present; during the night, the number of serious and minor accidents is relatively very low. As it has already been found before in the literature, e.g. [20], it is observed here that fatal accidents tend to take place in rural areas with a higher frequency and they are more evenly spread throughout the day. In addition, fewer accidents occur at night, except possibly on Friday and Saturday night. It is known that the characteristics of the rural crashes (more likely to be head-on crashes and single vehicle collisions with stationary objects [20]), rural drivers' demographics (larger proportion of older drivers in rural areas, with increased fragility and higher chances of dying [21]), their typical behaviours (their travelling speeds in rural areas may be greater [20], they are less likely to use seat belts [22], etc.) or the difficulty of obtaining timely medical assistance on rural roads [23,24], are contributing factors to the higher frequency of fatal accidents in rural than in urban areas. Given that it is estimated that approximately 20% of fatal accidents in E&W during 2017 were drink-drive accidents [25], the rise in the number of fatal accidents on Friday and Saturday nights might be due to the fact that alcohol consumption is higher on the weekends [26,27].

Turning the attention to urban areas, an approximate power-law scaling behaviour between the number of urban road accidents and the population size of the urban areas where they take place is assumed for E&W data. The total number of accidents at each location scales faster than linearly. This superlinear effect becomes stronger for the lowest degree of severity of the accidents, i.e. minor accidents are modelled via a power law whose scaling exponent has a higher value than for fatal and serious accidents. Generally speaking, urban areas in E&W do not present economies of scale in terms the number of road accidents.

Based on the 2008–2018 dataset of accidents in E&W, it is also observed that the probability of suffering at least one accident of any degree of severity in an urban area over the course of a year increases sublinearly with population size. However, the probability that a given accident is fatal or serious decreases sublinearly and the probability that it is minor increases sublinearly. That is, on a *per capita* basis, more populated urban areas in E&W are more prone to accidents, but these are less deadly than in the less populated counterparts.

But, why are these population scaling behaviours observed? It is a known fact that, as the population size of an urban settlement increases, the road surface also increases but it does so sublinearly [5]. It is, therefore, expected that, due to this reduction of space, the traffic congestion delay—which could perhaps be classified as an environmental factor—increases superlinearly, as shown in [28]. Whether traffic congestion has an impact on the frequency of road accidents still remains as an open question, due to the different conclusions reached by different studies. For example, [29] suggest that traffic congestion has little or no impact on the frequency of road accidents, although their results are constrained to the M25 orbital London motorway. Others conclude that congestion could lead to a reduction in the number of fatalities [30], but again, this result is restricted to highways and only considers fatal accidents. In [31,32], it is shown that variations in traffic significantly influence accident occurrence although they have a generally mixed influence on accident severity: low severity accidents tend to occur in congested traffic flow conditions, whereas severe and fatal accidents occur more often when the traffic is uncongested and when there are large differences in speed between adjacent lanes. These conclusions seem to agree with the result obtained here with regards to fatal accidents: the majority of fatal accidents occur in rural areas and the power-law relation with population size generally presents a lower scaling exponent than for accidents of minor severity.

Even if traffic congestion did not have a direct impact on the frequency of road accidents, previous works show that stress levels from drivers would be higher when driving in highly congested traffic conditions [33,34] and their satisfaction levels would be worse due to an increase in the travelling time [35]. Furthermore, drivers' stress levels are influenced not only by aspects related to the driving context—such as traffic congestion—but by a myriad of situational and personal factors [36] that seem to be enhanced in cities. Several studies [37–39] suggest that urban life could be related to certain mental health disorders, with anxiety, depression and socio-economic stress among them. A poor mental health status along with other psychological states related to sleep, fatigue, alertness, physical activity, emotional situation, etc. have been demonstrated to be a risk factor for road accidents [40–42]. Therefore, an increase in traffic congestion as the population size of an urban area gets larger contributes to the drivers' higher level of stress and, together with many other urban factors, could plausibly cause the observed superlinear scaling laws. A fully comprehensive analysis of the causes of the different scaling behaviours for accidents of different severity requires cross-disciplinary work from experts in urbanism, psychologists, engineers and policy-makers, since road accidents are the result of highly complex interactions of environmental, driver, vehicle, socio-economic and legislative factors.

It is important to note that notwithstanding its mathematical simplicity, scaling theory comes with some caveats. For example, the sensitivity analysis performed here (see §A.4), shows that the estimated values of scaling exponents fluctuate for choices of the minimum or the maximum population size. Indeed, it is known that scaling exponents fluctuate considerably when they are different from one, depending, for example, on how the boundaries of the different urban areas are defined [43]. The comparison between the results presented here for different types of area and different degrees of severity, can still prove to be insightful as long as all the assumptions about the data and the modelling techniques are taken into account for their interpretation.

In the light of the generally observed superlinear scaling behaviour of road accidents, especially those of minor severity, multiple ways of proceeding can be proposed. One is that urbanization and road accident prevention strategies should now be conceived with a special focus on population distribution among the different cities within a country. Another is that the creation of these strategies should keep the alleviation of the causes of urban road accidents as main goal.

# 5. Conclusion

Patterns related to the incidence of road accidents in E&W during the period 2008–2018 are explored. Firstly, time-series data is produced to assess the occurrence of accidents of different degrees of severity in both urban and rural areas. Some already known facts are verified: serious and minor accidents happen with higher frequency in urban than in rural areas and they follow daily patterns that reflect the working schedule of residents in E&W (two peaks of different height during weekdays, the short one corresponding to the morning rush hour and the tall one, to the evening rush hour; less accidents during the weekends and almost no accidents at nights). Fatal accidents happen more often in rural areas, perhaps due to the characteristics of rural crashes, the demographics of rural drivers, their typical behaviours or the higher difficulty to access medical care

after a crash. Fatal accidents in urban roads also tend to be more evenly spread throughout the day. At night, the number of fatal accidents is lower, possibly with the exception of Friday and Saturday nights.

Focusing on urban accidents, it is hypothesized that a power-law function describes the relationship between the number of accidents within the prescribed urban areas and the population size of those areas. The maximum-likelihood estimates for the parameters of this power law are obtained by applying a Poisson regression model. Accidents are classified as urban if they happen within the boundaries of BUAs that have a population larger than 10 000 (see §A.1), although other values for the minimum and maximum population size are also considered (see §A.4). It is observed that the values for the scaling exponents are generally greater than 1, although the values become closer to 1 for higher degrees of severity.

According to the 2008–2018 dataset for road accidents in E&W, the likelihood that an individual suffers at least one accident in an urban area over the course of a period of time increases sublinearly with population size. However, as the population increases, the likelihood that a given accident is fatal decreases sublinearly. So it is concluded that, in E&W, bigger conurbations pose a higher risk of accident but the accidents are less dangerous.

Traffic congestion, which also scales superlineary with the cities' population sizes, is postulated as an indirect cause of the observed behaviour, since it can lead to an increase in the drivers' stress levels, which together with other urban stressors, would lead to a larger number of accidents in urban areas of bigger population size. Owing to the rapid urbanization that the planet is undergoing, the results presented here have important implications for policy-makers as they seem to point towards a higher road accident rate in bigger cities.

Ethics. We adhere to the Royal Society Open Science ethics policy.

Data accessibility. The data that were used to obtain figures 2 and 3a,b is included as electronic supplementary material. These data are generated from the Road Safety datasets described in §2 of the manuscript, which is in turn used to obtain figure 1.

Authors' contributions. C.C.-A. contributed to the design of the study, the mathematical model, data collection and analysis of results. C.C.-A. and S.R.B. contributed to the interpretation of results and wrote the manuscript. R.P.C. contributed to the mathematical model and design of the manuscript.

Competing interests. We declare we have no competing interests.

Funding. R.P.C. receives financial support from the PEAK Urban programme, funded by UKRI's Global Challenge Research Fund, grant no. ref.: PES/P011055/1.

Acknowledgements. Running alongside his position at UCL, S.R.B. is also Fellow of the Alan Turing Institute.

# Appendix

## A.1. Defining urban areas

As pointed out by Batty & Ferguson in [44], due to the increasing opportunities for interactions enhanced by the thriving technologies and infrastructures, urban agglomerations can no longer be characterized by physical location, but by the set of interactions and flows that determine the urban network. However, a classification based on land use is deemed appropriate for road safety purposes, as the environmental features of roads play a very significant role in conditioning the occurrence of an accident. The BUAs in E&W were created by the Ordnance Survey for the 2011 UK Census following precisely this land use criterion. The digital boundaries of the BUAs are generated by firstly subdividing the territory in a 50 m grid based on the British National Grid System. Then, the percentage coverage of each of the four established classes of land use is determined for each cell, and finally, the cells that meet the minimum percentages to be considered of urban land use are grouped into the urban polygons that form the BUAs.

In this work, the accidents are classified as urban if (i) they are registered as urban in the Road Safety databases, and (ii) the registered lower super-output area (LSOA) of their location lies within a BUA with more than 10 000 people, according to the lookup table retrieved from [45]. The mid-year population estimate of each BUA is computed by summing over the mid-year population estimates published in [46] for all the LSOAs that lie within the BUA, again according to [45]. These mid-year population estimates corresponding to the years in the period 2008–2018 for each BUA can be found in the electronic supplementary material. The file also includes the population densities for each BUA in the year 2011 (taken from Table 'KS101EW—Usual resident population', retrieved from [47]), when the last UK Census took place. The accidents that occur outside a BUA or within BUAs of population size smaller than 10 000 are classified as rural. Hence, for the scaling analysis, only BUAs with at least 10 000 usual residents will

**Table 3.** Maximum-likelihood estimates for the scaling exponents corresponding to the power laws describing the relation between the number of accidents of different degrees of severity in urban areas of England and Wales and their population size in each individual year from 2008 to 2018.

| | $\beta_F$ | $\beta_S$ | $\beta_M$ | $\beta_T$ |
|---|---|---|---|---|
| 2008 | $1.076 \pm 0.018$ | $1.095 \pm 0.005$ | $1.087 \pm 0.002$ | $1.088 \pm 0.002$ |
| 2009 | $1.124 \pm 0.019$ | $1.089 \pm 0.005$ | $1.094 \pm 0.002$ | $1.094 \pm 0.002$ |
| 2010 | $1.083 \pm 0.022$ | $1.089 \pm 0.005$ | $1.107 \pm 0.002$ | $1.105 \pm 0.002$ |
| 2011 | $1.117 \pm 0.021$ | $1.068 \pm 0.005$ | $1.103 \pm 0.002$ | $1.098 \pm 0.002$ |
| 2012 | $1.096 \pm 0.021$ | $1.080 \pm 0.005$ | $1.101 \pm 0.002$ | $1.098 \pm 0.002$ |
| 2013 | $1.099 \pm 0.022$ | $1.052 \pm 0.005$ | $1.114 \pm 0.002$ | $1.106 \pm 0.002$ |
| 2014 | $1.092 \pm 0.026$ | $1.038 \pm 0.004$ | $1.130 \pm 0.002$ | $1.188 \pm 0.002$ |
| 2015 | $1.083 \pm 0.020$ | $1.037 \pm 0.005$ | $1.135 \pm 0.002$ | $1.122 \pm 0.002$ |
| 2016 | $1.059 \pm 0.020$ | $1.048 \pm 0.004$ | $1.148 \pm 0.002$ | $1.133 \pm 0.002$ |
| 2017 | $1.092 \pm 0.020$ | $1.109 \pm 0.004$ | $1.166 \pm 0.002$ | $1.156 \pm 0.002$ |
| 2018 | $1.031 \pm 0.019$ | $1.117 \pm 0.004$ | $1.163 \pm 0.002$ | $1.154 \pm 0.002$ |

be considered as the 'urban areas', although different cut-offs for the minimum and maximum population size are also explored in the sensitivity analysis presented in §A.4.

## A.2. Severity of an accident

Aspects related to the area can affect the severity of an accident, the travelling speed being one of the most determining factors (e.g. [31,48,49]). Since this work discriminates between two types of areas—urban and rural—where driving behaviours have very different characteristics, it is appropriate to consider accidents of different degrees of severity separately.

Different countries and even different regions within a country may have different definitions for the severity of an accident. In Great Britain, an accident's severity is that of the most severely injured casualty. Human casualties are officially classified as fatal, serious or slight. Throughout this work, however, accidents where the most severe injury is slight, will be referred to as 'minor accidents'. Fatal casualties are those where the sustained injuries cause death less than 30 days after the accident (confirmed suicides are excluded). Injuries where the victim needs to be detained in hospital as an 'in-patient' are deemed to be serious. Fractures, concussions, internal injuries, crushing, burns (excluding friction burns), severe cuts, severe general shock requiring medical treatment and injuries causing death 30 or more days after the accident are also considered to be serious even if hospitalization is not involved. Finally, slight injuries, such as a sprain (including neck whiplash injury), bruise, cut or minor shock, are those that have a minor character and sometimes they do not even require medical treatment [50].

## A.3. Scaling exponents estimates for each individual year

Table 3 shows the estimates for the scaling exponents computed with the road accident data corresponding to each year and the mid-year population estimates for the urban areas, also corresponding to each year. In general, it can be observed that (i) for each year, the scaling is also superlinear, and (ii) minor accidents tend to have a stronger superlinear behaviour than fatal or serious accidents.

## A.4. Sensitivity analysis of the scaling exponent

In this section, a sensitivity analysis is performed in order to test variations in the value of the scaling exponents when different subsets of urban areas are considered for the computation of their maximum-likelihood estimate.

The scaling exponents shown in figure 3a are obtained for the sets of urban areas that result from removing the $k$ elements with the largest population sizes. So, for $k = 1$, the data point corresponding to Greater London (BUA code: E34004707) is removed, as this is the urban area with the largest

population size (9 787 426 inhabitants); for $k = 2$, the two data points corresponding to both Greater London and Greater Manchester (BUA code: E34005054; 2 553 379 inhabitants) are removed, as these are the two urban areas with the largest population sizes. The scaling exponents are computed up to $k = 200$. A similar process is followed to compute the scaling exponents shown in figure 3b, but this time, the $k'$ elements with the smallest population sizes are removed, with $k' = 0, \ldots, 200$.

According to figure 3a, for the range of $k$ considered, the estimated scaling exponents undergo changes of up to 45% with respect to their value when all the urban areas are considered. The low number of fatal accidents makes the corresponding scaling exponent be the most sensitive to changes in the urban areas taken into account. The big fluctuations in the value of the exponents when large urban areas are removed from the dataset might respond to the 'dragon-king effect', pointed out in [43,51]. Figure 3b shows that changes in the values of the scaling exponents are less significant when urban areas with small population sizes are removed.

In general, it can be concluded that the scaling remains superlinear for all the sets of urban areas considered (all the values for the scaling exponents are above 1). It can also be seen that the nonlinear effect of scaling for minor accidents is in general greater than for serious accidents; for fatal accidents, the scaling behaviour fluctuates a lot more due to the scarcity of the data when the largest urban areas are removed.

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
