## [Reviewer comments · Royal Society Open Science]

Review History

RSOS-191739.R0 (Original submission)

Review form: Reviewer 1

Is the manuscript scientifically sound in its present form?

Yes

Are the interpretations and conclusions justified by the results?

Yes

Is the language acceptable?

Yes

Do you have any ethical concerns with this paper?

No

Have you any concerns about statistical analyses in this paper?

No

Recommendation?

Accept with minor revision (please list in comments)

Comments to the Author(s)

This paper deals with the estimation of the incidence of a road accident injury in urban areas in England and Wales. The paper is well written with an adequate literature review. The presented methodology, while simplistic, is scientifically sound and successfully conveyed to the reader. Finally, the conclusions are valid and well supported by the data.

Some issues should be addressed for the further improvement of the paper:

- The objective of the paper should be written more clearly in both the abstract and the introduction of the paper.
- In chapter 2a, the authors refer that data concern 2017. However, in the dataset provided in the supplementary files 2011 accident data are included. Please correct accordingly.
- The sub-chapter 1c does not offer extra value in the paper, while its title is not relevant as well. It is suggested that the related to paper information to be added in the sections 1a and 2a.
- The title of Table 2 should be corrected.

Review form: Reviewer 2

Is the manuscript scientifically sound in its present form?

Yes

Are the interpretations and conclusions justified by the results?

No

Is the language acceptable?

Yes

Do you have any ethical concerns with this paper?

No

Have you any concerns about statistical analyses in this paper?

Yes

Recommendation?

Major revision is needed (please make suggestions in comments)

Comments to the Author(s)

This paper presents an effort to link road accidents to socioeconomic sizes i.e. population and urbanization. In the field of road safety, this is not something new. On the contrary, several studies have taken into account such metrics, in combination with others, to explore their impact on the number and the severity of road accidents. Based on international literature, it is well established that the number of road accidents increases in urban areas due to the increased number of traffic conflicts. Moreover, the severity of accidents increases in rural areas due to the higher travel speeds.

The specific paper mathematically verifies existing road safety knowledge. Thus, it could be more appealing to mathematicians than to road safety scientists. It remains to the journal Editor to decide whether or not this paper is suitable for publication in the specific journal.

In any case, the following points should be taken into consideration for the overall improvement of the paper:

- In part (a) of the introduction, several statements are made without indicating the respective source. E.g. in the first paragraph a WHO reference is included only in the first sentence. Please indicate all used sources in the text and in the reference list. If several sentences (e.g. a whole

paragraph) are based on the same source, please add it in more than one places (e.g. at the beginning and at the end of the paragraph).

- In part (c) of the introduction, it is written "... the UK's participation in projects such as Vision Zero ...". Vision Zero is not a project but a road safety strategy, which initiated in Sweden in the 90's and since then, has been adopted by numerous countries in the framework of the Safe System approach towards road safety. Please adjust the text accordingly.

- Data used in this paper refer to only one year (2017). This is not common in road safety research as the number of road accidents and casualties in one year may be strongly affected by local, temporary conditions (e.g. existing work zones, infrastructure projects, traffic arrangements or even one major accident with multiple casualties). Therefore, time-series analysis should use data from an at least 10-year period. Especially in the UK, where complete and reliable road safety data are available for several decades, this is possible and necessary.

- According to the text, data used refer to 2017. However, in the supplementary excel file available with this submission, it is written 2011. Please, clarify and correct as necessary. In addition, the data used in this work are not accessible through the website indicated in the text (data.gov.uk) (at least it is not clear how they could be accessed). Please provide the specific link to the used data.

In addition, in the excel file, it is not clear what "Total" means in column titled "Rural urban". Please explain this.

- The text in Data and methods (b) presents a light repetition in comparison to (a). Please amend it to avoid that.

- The authors suggest that the increased number of fatalities in rural areas is due to the higher speed limits there. Indeed speed limits are the legislative "official" measure of behaviour however, road safety research has indicated that what is even more important are adopted / travelling speeds and these may differ significantly from speed limits. Thus, it is more appropriate to discuss about travel speeds than speed limits.

Decision letter (RSOS-191739.R0)

02-Jan-2020

Dear Professor Cabrera-Arnau,

The editors assigned to your paper ("Uncovering the behaviour of road accidents in urban areas") have now received comments from reviewers. We would like you to revise your paper in accordance with the referee suggestions which can be found below (not including confidential reports to the Editor). Please note this decision does not guarantee eventual acceptance.

Please submit a copy of your revised paper before 25-Jan-2020. Please note that the revision deadline will expire at 00.00am on this date. If we do not hear from you within this time then it will be assumed that the paper has been withdrawn. In exceptional circumstances, extensions may be possible if agreed with the Editorial Office in advance. We do not allow multiple rounds of revision so we urge you to make every effort to fully address all of the comments at this stage. If deemed necessary by the Editors, your manuscript will be sent back to one or more of the original reviewers for assessment. If the original reviewers are not available, we may invite new reviewers.

- Data accessibility

If you wish to submit your supporting data or code to Dryad (<http://datadryad.org/>), or modify your current submission to dryad, please use the following link:
<http://datadryad.org/submit?journalID=RSOS&manu=RSOS-191739>

- Competing interests

- Authors' contributions

- Acknowledgements

- Funding statement

on behalf of Professor Zhong-Ke Gao (Associate Editor) and Mark Chaplain (Subject Editor)
openscience@royalsociety.org

Comments to Author:

Reviewers' Comments to Author:

Reviewer: 1

Comments to the Author(s)

This paper deals with the estimation of the incidence of a road accident injury in urban areas in England and Wales. The paper is well written with an adequate literature review. The presented methodology, while simplistic, is scientifically sound and successfully conveyed to the reader. Finally, the conclusions are valid and well supported by the data.

Some issues should be addressed for the further improvement of the paper:

- The objective of the paper should be written more clearly in both the abstract and the introduction of the paper.
- In chapter 2a, the authors refer that data concern 2017. However, in the dataset provided in the supplementary files 2011 accident data are included. Please correct accordingly.
- The sub-chapter 1c does not offer extra value in the paper, while its title is not relevant as well. It is suggested that the related to paper information to be added in the sections 1a and 2a.
- The title of Table 2 should be corrected.

Reviewer: 2

Comments to the Author(s)

This paper presents an effort to link road accidents to socioeconomic sizes i.e. population and urbanization. In the field of road safety, this is not something new. On the contrary, several studies have taken into account such metrics, in combination with others, to explore their impact on the number and the severity of road accidents. Based on international literature, it is well established that the number of road accidents increases in urban areas due to the increased number of traffic conflicts. Moreover, the severity of accidents increases in rural areas due to the higher travel speeds.

The specific paper mathematically verifies existing road safety knowledge. Thus, it could be more appealing to mathematicians than to road safety scientists. It remains to the journal Editor to decide whether or not this paper is suitable for publication in the specific journal.

In any case, the following points should be taken into consideration for the overall improvement of the paper:

- In part (a) of the introduction, several statements are made without indicating the respective source. E.g. in the first paragraph a WHO reference is included only in the first sentence. Please indicate all used sources in the text and in the reference list. If several sentences (e.g. a whole

paragraph) are based on the same source, please add it in more than one places (e.g. at the beginning and at the end of the paragraph).

- In part (c) of the introduction, it is written "... the UK's participation in projects such as Vision Zero ...". Vision Zero is not a project but a road safety strategy, which initiated in Sweden in the 90's and since then, has been adopted by numerous countries in the framework of the Safe System approach towards road safety. Please adjust the text accordingly.

- Data used in this paper refer to only one year (2017). This is not common in road safety research as the number of road accidents and casualties in one year may be strongly affected by local, temporary conditions (e.g. existing work zones, infrastructure projects, traffic arrangements or even one major accident with multiple casualties). Therefore, time-series analysis should use data from an at least 10-year period. Especially in the UK, where complete and reliable road safety data are available for several decades, this is possible and necessary.

- According to the text, data used refer to 2017. However, in the supplementary excel file available with this submission, it is written 2011. Please, clarify and correct as necessary. In addition, the data used in this work are not accessible through the website indicated in the text (data.gov.uk) (at least it is not clear how they could be accessed). Please provide the specific link to the used data.

In addition, in the excel file, it is not clear what "Total" means in column titled "Rural urban". Please explain this.

- The text in Data and methods (b) presents a light repetition in comparison to (a). Please amend it to avoid that.

- The authors suggest that the increased number of fatalities in rural areas is due to the higher speed limits there. Indeed speed limits are the legislative "official" measure of behaviour however, road safety research has indicated that what is even more important are adopted / travelling speeds and these may differ significantly from speed limits. Thus, it is more appropriate to discuss about travel speeds than speed limits.

Author's Response to Decision Letter for (RSOS-191739.R0)

See Appendix A.

RSOS-191739.R1 (Revision)

Review form: Reviewer 1

Is the manuscript scientifically sound in its present form?

Yes

Are the interpretations and conclusions justified by the results?

Yes

Is the language acceptable?

Yes

Do you have any ethical concerns with this paper?

No

Have you any concerns about statistical analyses in this paper?

No

Recommendation?

Accept with minor revision (please list in comments)

Comments to the Author(s)

This is a revised paper dealing with the estimation of the incidence of a road accident injury in urban areas in England and Wales. The authors have addressed successfully all comments. A minor issue is suggested to be tackled before the publication of the paper:

- The adoption of the vision zero by the UK is not relevant with the description of the accident data in the section 2a. The authors are suggested either to add this paragraph in the introduction or delete it.

Review form: Reviewer 2

Is the manuscript scientifically sound in its present form?

Yes

Are the interpretations and conclusions justified by the results?

Yes

Is the language acceptable?

Yes

Do you have any ethical concerns with this paper?

No

Have you any concerns about statistical analyses in this paper?

No

Recommendation?

Accept as is

Comments to the Author(s)

The current version of the paper shows a significant effort on behalf of the authors to address all reviewers' comments.

The objective and the intended audience of this work are now clearer. All in all, the paper is now more cohesive and comprehensive and it is considered appropriate for publication.

One last suggestion to the authors is to use the following link as a reference to Vision Zero:

<https://www.trafikverket.se/en/startpage/operations/Operations-road/vision-zero-academy/>

The link currently included in the text does not refer to the initial concept of Vision Zero but to the Vision Zero Network which is a collaborative campaign helping communities reach their goals of Vision Zero.

Decision letter (RSOS-191739.R1)

06-Mar-2020

Dear Professor Cabrera-Arnau:

On behalf of the Editors, I am pleased to inform you that your Manuscript RSOS-191739.R1

entitled "Uncovering the behaviour of road accidents in urban areas" has been accepted for publication in Royal Society Open Science subject to minor revision in accordance with the referee suggestions. Please find the referees' comments at the end of this email.

The reviewers and Subject Editor have recommended publication, but also suggest some minor revisions to your manuscript. Therefore, I invite you to respond to the comments and revise your manuscript.

- Ethics statement

- Data accessibility

If you wish to submit your supporting data or code to Dryad (<http://datadryad.org/>), or modify your current submission to dryad, please use the following link:
<http://datadryad.org/submit?journalID=RSOS&manu=RSOS-191739.R1>

- Competing interests

- Authors' contributions

- Acknowledgements

- Funding statement

Because the schedule for publication is very tight, it is a condition of publication that you submit the revised version of your manuscript before 15-Mar-2020. Please note that the revision deadline will expire at 00.00am on this date. If you do not think you will be able to meet this date please let me know immediately.

on behalf of Professor Zhong-Ke Gao (Associate Editor) and Mark Chaplain (Subject Editor)
openscience@royalsociety.org

Reviewer comments to Author:

Reviewer: 1

Comments to the Author(s)

This is a revised paper dealing with the estimation of the incidence of a road accident injury in urban areas in England and Wales. The authors have addressed successfully all comments. A minor issue is suggested to be tackled before the publication of the paper:

- The adoption of the vision zero by the UK is not relevant with the description of the accident data in the section 2a. The authors are suggested either to add this paragraph in the introduction or delete it.

Reviewer: 2

Comments to the Author(s)

The current version of the paper shows a significant effort on behalf of the authors to address all reviewers' comments.

The objective and the intended audience of this work are now clearer. All in all, the paper is now more cohesive and comprehensive and it is considered appropriate for publication.

One last suggestion to the authors is to use the following link as a reference to Vision Zero:

<https://www.trafikverket.se/en/startpage/operations/Operations-road/vision-zero-academy/>
The link currently included in the text does not refer to the initial concept of Vision Zero but to the Vision Zero Network which is a collaborative campaign helping communities reach their goals of Vision Zero.

Author's Response to Decision Letter for (RSOS-191739.R1)

See Appendix B.

Decision letter (RSOS-191739.R2)

19-Mar-2020

Dear Professor Cabrera-Arnau,

It is a pleasure to accept your manuscript entitled "Uncovering the behaviour of road accidents in urban areas" in its current form for publication in Royal Society Open Science.

You can expect to receive a proof of your article in the near future. Please contact the editorial office (openscience_proofs@royalsociety.org) and the production office (openscience@royalsociety.org) to let us know if you are likely to be away from e-mail contact -- if

you are going to be away, please nominate a co-author (if available) to manage the proofing process, and ensure they are copied into your email to the journal.

Kind regards,

on behalf of Professor Zhong-Ke Gao (Associate Editor) and Mark Chaplain (Subject Editor)
openscience@royalsociety.org

Appendix A

Response to Reviewers

We thank the reviewers for the time to analyse our submission. Based on their comments, we have made some amendments to the manuscript. In the following lines, we explain the specific action taken in response of the reviewers' comments.

REVIEWER 1

Reviewer (R): *This paper deals with the estimation of the incidence of a road accident injury in urban areas in England and Wales. The paper is well written with an adequate literature review. The presented methodology, while simplistic, is scientifically sound and successfully conveyed to the reader. Finally, the conclusions are valid and well supported by the data.*

Some issues should be addressed for the further improvement of the paper:

The objective of the paper should be written more clearly in both the abstract and the introduction of the paper.

Authors (A): Thank you for the observation. The objectives have been presented more clearly in both the abstract (page 1) and the introduction (page 2).

R: *In chapter 2a, the authors refer that data concern 2017. However, in the dataset provided in the supplementary files 2011 accident data are included. Please correct accordingly.*

A: Thank you for pointing that out, we made a mistake when naming the supplementary file. The files have now the correct headings. Moreover, more files have now been added related to the years 2008-2018, as per the other reviewer's recommendation of exploring data corresponding to a longer period.

R: *The sub-chapter 1c does not offer extra value in the paper, while its title is not relevant as well. It is suggested that the information related to paper is added in the sections 1a and 2a.*

A: Section 1c has been removed. The relevant contents have been included in sections 1a and 2a.

R: *The title of Table 2 should be corrected.*

A: The typo has now been amended, many thanks.

REVIEWER 2

R: *This paper presents an effort to link road accidents to socioeconomic sizes i.e. population and urbanization. In the field of road safety, this is not something new. On the contrary, several studies have taken into account such metrics, in combination with others, to explore their impact on the number and the severity of road accidents. Based on international literature, it is well established that the number of road accidents increases in urban areas due to the increased number of traffic conflicts. Moreover, the severity of accidents increases in rural areas due to the higher travel speeds.*

The specific paper mathematically verifies existing road safety knowledge. Thus, it could be more appealing to mathematicians than to road safety scientists. It remains to the journal Editor to decide whether or not this paper is suitable for publication in the specific journal.

A: The abstract (page 1) and introduction (last two paragraphs of section 1a and last paragraph of section 1b; pages 2 and 3) have been rephrased to express the aims and value of this paper with greater clarity. In the first paragraph of the discussion (page 8) and the first paragraph of the conclusion (page 9), the authors now acknowledge that, indeed, some of the results presented in the paper have been already explored by previous researchers.

R: *In any case, the following points should be taken into consideration for the overall improvement of the paper: in part (a) of the introduction, several statements are made without indicating the respective source. E.g. in the first paragraph a WHO reference is included only in the first sentence. Please indicate all used sources in the text and in the reference list. If several sentences (e.g. a whole paragraph) are based on the same source, please add it in more than one places (e.g. at the beginning and at the end of the paragraph).*

A: The locations of the citations have been updated following the suggestions given in the reviewer's comment.

R: *In part (c) of the introduction, it is written "... the UK's participation in projects such as Vision Zero ...". Vision Zero is not a project but a road safety strategy, which initiated in Sweden in the 90's and since then, has been adopted by numerous countries in the framework of the Safe System approach towards road safety. Please adjust the text accordingly.*

A: Thank you for the clarification. This information has been incorporated and it is now located in section 2a (page 3), according to the other reviewer's suggestion.

R: *Data used in this paper refer to only one year (2017). This is not common in road safety research as the number of road accidents and casualties in one year may be strongly affected by local, temporary conditions (e.g. existing work zones, infrastructure projects, traffic arrangements or even one major accident with multiple casualties). Therefore, time-series analysis should use data from an at least 10-year period. Especially in the UK, where complete and reliable road safety data are available for several decades, this is possible and necessary.*

A: The authors appreciate the need to use data from a longer period. The results for the time-series analysis have now been obtained for the time period spanning from the year 2008 to the year 2018 (since the first submission, the dataset corresponding to 2018 was released). Similarly, the results for the scaling analysis have been recomputed with the larger data set; the values of the scaling exponents for each individual year have also been included in the

Appendix (page 12). The manuscript has been modified throughout to account for the change of dataset.

R: According to the text, data used refer to 2017. However, in the supplementary excel file available with this submission, it is written 2011. Please, clarify and correct as necessary. In addition, the data used in this work are not accessible through the website indicated in the text (data.gov.uk) (at least it is not clear how they could be accessed). Please provide the specific link to the used data.

A: The typo has now been amended and the link has been replaced (page 4).

R: In addition, in the excel file, it is not clear what “Total” means in the column titled “Rural urban”. Please explain this.

A: Total refers to the total number of accidents across different degrees of severity. The word “Total” has been substituted by “All severities”.

R: The text in Data and methods (b) presents a light repetition in comparison to (a). Please amend it to avoid that.

A: Thank you for the suggestion, this has been rephrased accordingly.

R: The authors suggest that the increased number of fatalities in rural areas is due to the higher speed limits there. Indeed speed limits are the legislative “official” measure of behaviour however, road safety research has indicated that what is even more important are adopted / travelling speeds and these may differ significantly from speed limits. Thus, it is more appropriate to discuss about travel speeds than speed limits.

A: Thank you for this observation. The text in the discussion (pages 8 and 9) and the conclusion (pages 9 and 10) has been modified. Other possible causes for the increased number of fatalities in rural areas have also been discussed and the following references, corresponding to numbers 21-25, have been added to amplify this discussion:

Zwerling, C., Peek-Asa, C., Whitten, P.S., Choi, S-W., Sprince, N.L., Jones, M.P. (2005). Fatal motor vehicle crashes in rural and urban areas: decomposing rates into contributing factors. *Injury Prevention*.

Tefft, B. C (2017). Rates of Motor Vehicle Crashes, Injuries and Deaths in Relation to Driver Age, United States, 2014-2015. *AAA Foundation for Traffic Safety*.

Beck, L.F., Downs, J., Stevens, M.R., Sauber-Schatz, E.K. (2017). Rural and Urban Differences in Passenger-Vehicle–Occupant Deaths and Seat Belt Use Among Adults — United States, 2014. *MMWR Surveillance Summaries*.

Gonzalez, R., Cummings, G., Mulekar, M., Rodning, C.B. (2006). Increased Mortality in Rural Vehicular Trauma: Identifying Contributing Factors Through Data Linkage. *The Journal of Trauma: Injury, Infection, and Critical Care*.

Byrne, J. P., Mann, N. C., Dai, M., Mason, S. A., Karanicolas, P., Rizoli, S., Nathens, A. B. (2019). Association Between Emergency Medical Service Response Time and Motor Vehicle Crash Mortality in the United States. *JAMA Surgery*.

Appendix B

Response to Decision Letter

We thank once again the editors and reviewers for their time to analyse our submission. We have now included all the end statements in our article as recommended by the editors. In the following lines, we explain the specific action taken in response of the reviewers' comments.

REVIEWER 1

Reviewer (R): *This is a revised paper dealing with the estimation of the incidence of a road accident injury in urban areas in England and Wales. The authors have addressed successfully all comments. A minor issue is suggested to be tackled before the publication of the paper:*

- The adoption of the vision zero by the UK is not relevant with the description of the accident data in the section 2a. The authors are suggested either to add this paragraph in the introduction or delete it.

Authors (A): Thank you very much for the suggestion. We decided to include this information in the introduction. According to the other reviewer's comment, we also updated the hyperlink to Vision Zero.

REVIEWER 2

R: *The objective and the intended audience of this work are now clearer. All in all, the paper is now more cohesive and comprehensive and it is considered appropriate for publication. One last suggestion to the authors is to use the following link as a reference to Vision Zero: <https://www.trafikverket.se/en/startpage/operations/Operations-road/vision-zero-academy/>. The link currently included in the text does not refer to the initial concept of Vision Zero but to the Vision Zero Network which is a collaborative campaign helping communities reach their goals of Vision Zero.*

A: Thank you for the remark. We have now updated the link and, according to the other reviewer's suggestion, we have placed the information about Vision Zero in the introduction.